# Exploring the roles of RNAs in chromatin architecture using deep learning

**Shuzhen Kuang** [1] **& Katherine S. Pollard** [1,2,3] ✉

Recent studies have highlighted the impact of both transcription and transcripts on 3D genome organization, particularly its dynamics. Here, we propose a deep learning framework, called AkitaR, that leverages both genome sequences and genome-wide RNA-DNA interactions to investigate the roles of chromatin-associated RNAs (caRNAs) on genome folding in HFFc6 cells. In order to disentangle the *cis-* and *trans-*regulatory roles of caRNAs, we have compared models with nascent transcripts, *trans-*located caRNAs, open chromatin data, or DNA sequence alone. Both nascent transcripts and *trans-*located caRNAs improve the models' predictions, especially at cell-type-specific genomic regions. Analyses of feature importance scores reveal the contribution of caRNAs at TAD boundaries, chromatin loops and nuclear substructures such as nuclear speckles and nucleoli to the models' predictions. Furthermore, we identify non-coding RNAs (ncRNAs) known to regulate chromatin structures, such as *MALAT1* and *NEAT1*, as well as several new RNAs, *RNY5*, *RPPH1*, *POLG-DT* and *THBS1-IT1*, that might modulate chromatin architecture through *trans-*interactions in HFFc6. Our modeling also suggests that transcripts from Alus and other repetitive elements may facilitate chromatin interactions through *trans* R-loop formation. Our findings provide insights and generate testable hypotheses about the roles of caRNAs in shaping chromatin organization.

The human genome is folded into complex structures within the nucleus with multiple levels of organization, including compartments, topologically associated domains (TADs) and chromatin loops[1,2]. This spatial organization is dynamic and varies across cell types and tissues, and it is interconnected with cellular processes such as gene transcription and DNA replication[3–5]. Recent studies have unraveled the critical roles of CTCF and cohesin in three-dimensional (3D) genome organization, including their involvement in TAD and loop formation via the loop extrusion mechanism[6–8]. Other proteins, such as YY1 and ZNF143, are potentially also regulating chromatin organization[9–12]. However, all these structural proteins are widely expressed, and alone cannot explain the dynamic and cell-type specific aspects of chromatin organization.

A growing number of studies point to transcription as a potential contributor to the dynamic aspects of genome folding[13–17]. While 3D chromatin structures are known to play a role in gene silencing and activation, the process of transcription can in turn affect 3D genome folding in a cell-type- or tissue-specific manner[13,18,19]. For example, TAD boundaries are often located near or at active gene promoters[3]. Furthermore, transcribing RNA polymerases (RNAPs) are reported to act as moving barriers for the loop-extruding cohesins[13]. Thus, some chromatin dynamics are expected to reflect a *cis* effect of nascent transcription.

Transcribed RNA molecules may also contribute to chromatin dynamics. Specifically, RNAs known as chromatin-associated RNAs (caRNAs) have been observed to directly interact with DNA or to bind

[1]Gladstone Institute of Data Science and Biotechnology, San Francisco, CA, USA. [2]Department of Epidemiology & Biostatistics, University of California, San Francisco, CA, USA. [3]Chan Zuckerberg Biohub, San Francisco, CA, USA. ✉e-mail: katherine.pollard@gladstone.ucsf.edu

chromatin-associated proteins[14,16,20,21]. These caRNAs include nascent RNAs, long non-coding RNAs (lncRNAs), small nuclear RNAs (snRNAs), small nucleolar RNAs (snoRNAs), enhancer RNAs (eRNAs) and repeat RNAs[15–17,22,23]. Most caRNAs bind close to their locus of origin (*cis*-interactions), but many interact with distant genomic loci (*trans*-interactions). Several of the latter *trans*-located caRNAs have been implicated in chromatin regulation. For example, lncRNA *HOTTIP* promotes distal TAD formation by forming RNA-DNA hybrid structures (R-loop) in leukemia[24]. Enhancer RNAs have been proposed to mediate promoter-enhancer interactions by forming *trans* R-loops at Alu sequences[25]. Several other *trans*-located caRNAs, such as lncRNAs *MALAT1*, *NEAT1*, and *Firre*, also play critical roles in chromatin organization[26–29].

CaRNAs, particularly non-coding RNAs (ncRNAs), are proposed to shape 3D genome structure via multiple mechanisms[14–17]. First, caRNAs can recruit chromatin regulatory proteins to specific genomic loci. For example, caRNAs have been found to directly bind CTCF and serve as locus-specific factors to recruit CTCF to TAD boundaries and loop anchors[24,30–34]. Perturbing the abundance of RNAs or mutating the RNA-binding region of CTCF weakens the insulation of TAD boundaries or disrupts the formation of chromatin loops[24,30,31,34]. Second, caRNAs can act as scaffolds to organize chromosomal architecture by integrating multiple regulatory proteins. A well-known example is the lncRNA *Xist*, which initiates and maintains X chromosome inactivation by interacting with proteins[15]. Third, caRNAs can drive phase separation and coordinate the formation of various membrane-less nuclear bodies[16,17]. For example, the lncRNA *NEAT1* induces the assembly of paraspeckles via phase separation and is indispensable for this nuclear structure[26,35].

Given the relatively small number of functionally characterized *trans*-located caRNAs in genome folding, we hypothesized that other examples remain to be discovered. To explore this hypothesis, we used machine learning and bioinformatics tools to interrogate RNA-DNA interaction data. Several high-throughput approaches have been developed to globally profile caRNAs, including chromatin-associated RNA sequencing (ChAR-seq)[36], global RNA interaction with DNA sequencing (GRID-seq)[37], RNA & DNA split-pool recognition of interactions by tag extension (RD-SPRITE)[28], mapping of RNA-genome interactions (MARGI)[21] and its improved version in-situ MARGI (iMARGI)[20,38,39]. These techniques enable genome-scale investigations of the mechanisms through which dynamically expressed caRNAs contribute to nuclear organization.

Modeling 3D genome folding using machine learning offers an efficient way to study chromatin dynamics that complements experimental strategies. Recently, deep learning models, such as Akita[40], DeepC[41] and ORCA[42], have been developed to predict 3D genome structure from DNA sequence. Since these models are highly accurate, they enable researchers to decode sequence determinants of genome folding through computational techniques such as in silico mutagenesis and feature importance scores[43]. More recently, models incorporating epigenomic data were built to achieve cell-type-specific predictions[44,45]. Significantly, these models learned the sequence and epigenetic correlates of 3D genome folding. The capability of deep learning models to probe sequence and epigenetic dependencies of genome folding motivated us to use this approach to explore the roles of caRNAs in 3D genome architecture.

In this study, we thus extended the Akita model to predict cell-type-specific chromatin contact frequencies using not only DNA sequence but also RNA-DNA interaction data. We call the resulting modeling framework AkitaR. To advance our understanding of the *cis*- and *trans*-regulatory roles of caRNAs in chromatin architecture, we designed AkitaR to use either nascent RNA or *trans*-located caRNA. Comparisons of these models to each other and to models trained on sequence or open chromatin data allowed us to dissect how each of

these relate to chromatin interaction frequencies genome-wide. We showed that AkitaR achieved significantly better predictions on regions of the human genome with cell-type-specific genome folding. Particularly, some chromatin interactions were uniquely captured by the model with *trans*-located caRNAs. Examination of the feature importance scores showed not only the general contribution of caRNAs at CTCF peaks, TAD boundaries and loop anchors but also revealed slightly different contributions of different types of caRNAs at nuclear structures, such as snoRNAs in nucleoli and nuclear speckles. This enabled us to develop testable hypotheses about the roles of specific types of caRNAs in genome folding.

## Results
In order to characterize the roles of caRNAs in 3D genome folding, we downloaded the genome-wide RNA-chromatin interactions in human foreskin fibroblast cells (HFFc6) and human embryonic stem cells (H1ESC) captured by iMARGI and the corresponding genome-wide DNA-DNA interactions captured by Micro-C from 4DN data portal (https://data.4dnucleome.org/) (Supplementary Table 1)[20,46,47]. We chose iMARGI data over other techniques that map genome-wide RNA-DNA contacts, because iMARGI has been performed in human cell lines that have rich transcriptomic and epigenomic datasets we could use to interpret the high-quality chromatin interaction data. We used HFFc6 for our primary analyses, and leveraged H1ESC for identifying cell-type differences.

To disentangle the roles of nascent transcription versus *trans*-located caRNAs, both of which are involved in 3D genome organization, we broke down the human genome into 2048-bp bins and defined nascent transcripts as all the RNAs transcribed from a given 2048-bp DNA bin and *trans*-located caRNAs as all RNAs transcribed from at least 1 Mb away from the bin. We opted for 1 Mb to identify *trans*-located caRNAs instead of the 100 or 200 Kb used in previous studies[39,48] in order to align with the window size of our predictive models and also to remove self-interactions for the vast majority of genes (~99.9%). As different types of caRNAs may engage in different *trans*-interactions (Fig. 1a) and contribute to different chromatin features, we further classified *trans*-located caRNAs into eight groups: snRNAs, snoRNAs, other small RNAs, lncRNAs, misc_RNAs, RNAs from protein coding genes, RNAs from other types of genes and RNAs from regions without known gene annotation.

### CaRNAs interact with *cis* and *trans* located open chromatin
To explore how caRNAs are spatially localized inside the nucleus, we examined whether caRNAs identified by iMARGI preferentially interact with any parts of the genome. Similar to DNA-DNA interactions, we observed that RNAs tend to interact with DNA regions that share the same spatial or functional annotations as the loci from which they are transcribed, such as being in the same compartment or having the same Spatial Position Inference of the Nuclear genome (SPIN) state[49] or chromatin state identified by chromHMM[50,51] (Fig. 1b and Supplementary Fig. 1). Beyond that, caRNAs interact more frequently with DNA regions with high versus low transcriptional activity (Fig. 1b and Supplementary Fig. 1). This trend is confirmed by the enrichment of caRNAs at open chromatin regions (Fig. 1c). Interestingly, the enrichment was also observed for *trans*-located caRNAs (Fig. 1c), and the amount of *trans*-located caRNA attached to DNA regions was positively correlated with the region's chromatin accessibility (Pearson's R = 0.37).

Considering that many RNA-DNA interactions across spatial or functional annotations may be from *trans*-interactions, we assessed the percentage of RNA-DNA interactions occurring in *trans* in HFFc6 both globally and for each annotated gene. We included caRNAs transcribed from the DNA loci on the same chromosome but at least 1 Mb away plus those encoded on different chromosomes. CaRNAs

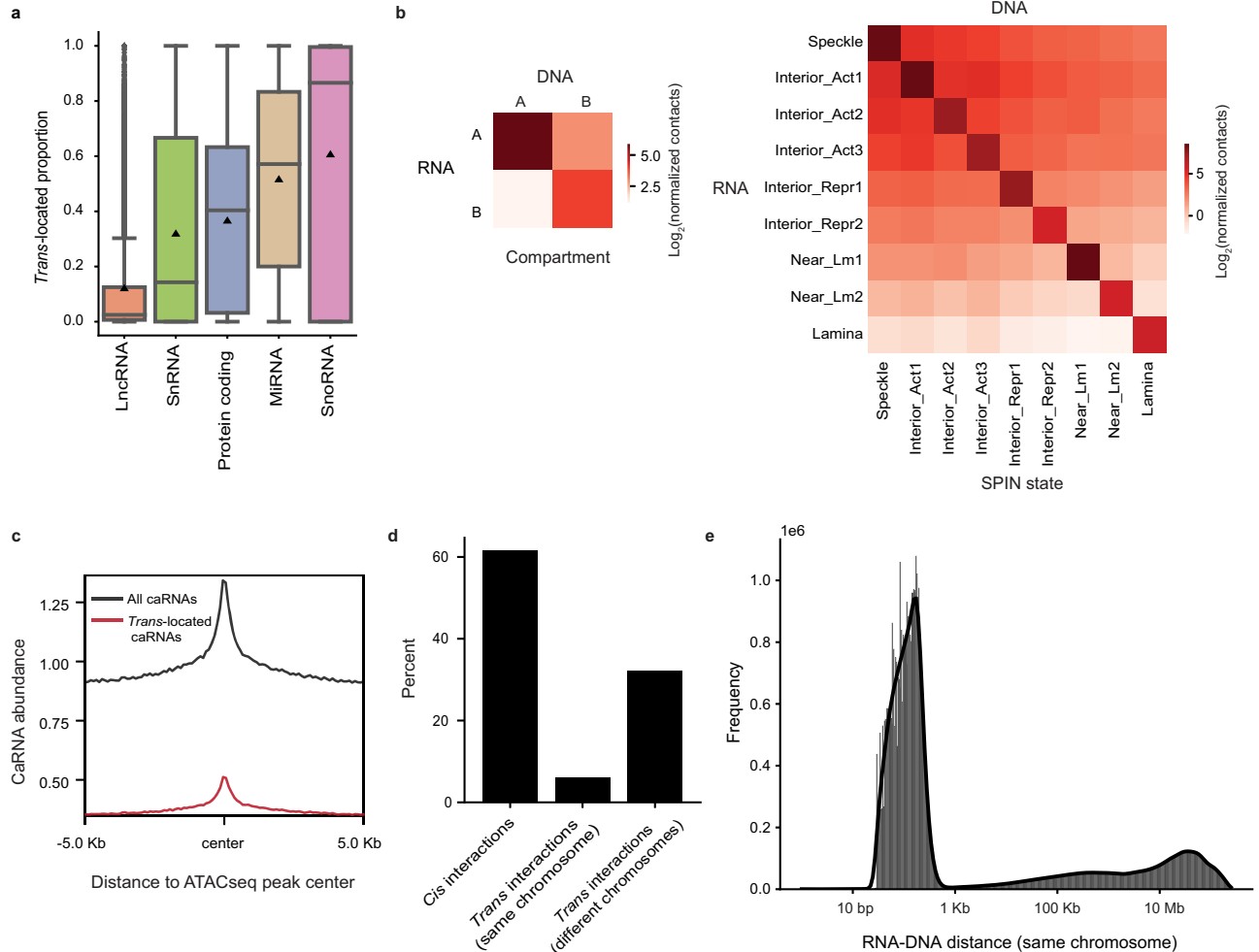

**Fig. 1 | Chromatin-associated RNAs preferentially bind open chromatin and many interactions occur in *trans*. a** The proportion of *trans*-interactions for RNAs transcribed from each gene that is annotated as lncRNA (*n* = 17,673), snRNA (*n* = 727), protein-coding (*n* = 19,317), miRNA (*n* = 900) or snoRNA (*n* = 765). The center line and triangle within the box represent the median and mean value, respectively. The box represents the interquartile range (IQR), with whiskers set to 1.5 times the IQR. Outliers are shown in points. **b** The number of RNA-DNA inter-actions (log₂ normalized count) within and across compartments (left panel) and SPIN states (right panel). The interaction frequencies were normalized to the size of compartments and SPIN states. **c** The abundance of all caRNAs or *trans*-located caRNAs at ATAC-seq peaks and their flanking regions. **d** The percentage of genome-wide *cis*-interactions (RNA-DNA distance 1 Mb or less) and *trans*-interactions (RNA-DNA distance > 1 Mb on the same chromosome or RNA and DNA on different chromosomes). **e** Histogram of RNA-DNA interaction frequencies as a function of genomic distance between DNA and RNA loci on the same chromosome. SPIN: Spatial Position Inference of the Nuclear genome, Interior_Act 1: Interior Active 1, Interior_Act 2: Interior Active 2, Interior_Act 3: Interior Active 3, Interior_Repr1: Interior Repressive 1, Interior_Repr2: Interior Repressive 2, Near_Lm1: Near Lamina 1, Near_Lm2: Near Lamina 2. Source data are provided as a Source Data file.

primarily interacted with proximal DNA regions (Fig. 1d), and of all the interactions on the same chromosome, over 90% spanned a distance of less than 1 Kb (Fig. 1e). Nevertheless, 38.38% of RNA-DNA interac-tions occurred in *trans*, including 6.14% within the same chromosome and 32.24% on different chromosomes (Fig. 1d). These results are quite different from DNA-DNA interactions from the Micro-C data, where *trans*-interactions across chromosome are less frequent (6.46% within the same chromosome and 11.92% on different chromosomes). This difference suggests that the proximity of most caRNAs to chromatin is not due to their being transcribed from DNA that is nearby in the 3D nucleus.

Notably, we observed that the majority of the small ncRNAs and a number of lncRNAs and RNAs from protein-coding genes were engaged in *trans*-interactions (Fig. 1a and Supplementary Fig. 2). Given the well-established importance of several snRNAs and snoRNAs in nuclear structures, these results suggest that other ncRNAs and tran-scripts of some protein-coding genes may also regulate chromatin structures.

## *Trans*-located caRNAs are particularly enriched at TAD boundaries

To investigate whether caRNAs play roles at particular landmarks within the 3D genome, we first used the iMARGI data to examine their abundance at TAD boundaries. Since most TAD boundaries are located in compartment A in HFFc6 (Fig. 2a) and tend to have higher chromatin accessibility compared to surrounding regions (Fig. 2b), we hypothe-sized that caRNAs would be enriched at TAD boundaries. In order to check whether nascent transcripts and *trans*-located RNAs follow similar patterns, we conducted separate analyses for each of them. As anticipated, *trans*-located caRNAs peaked at TAD boundaries and greatly decreased in flanking regions (± 50 Kb) (Fig. 2b). After cate-gorizing TAD boundaries in HFFc6 and H1ESC based on their strength and cell type specificity (see Methods, Fig. 2b), we found that HFFc6 *trans*-located caRNAs are significantly less prevalent at TAD bound-aries unique to H1ESC or with higher insulation strength in H1ESC than at TAD boundaries shared with or more prominent in HFFc6. Similar but weaker patterns were also observed for open chromatin signals

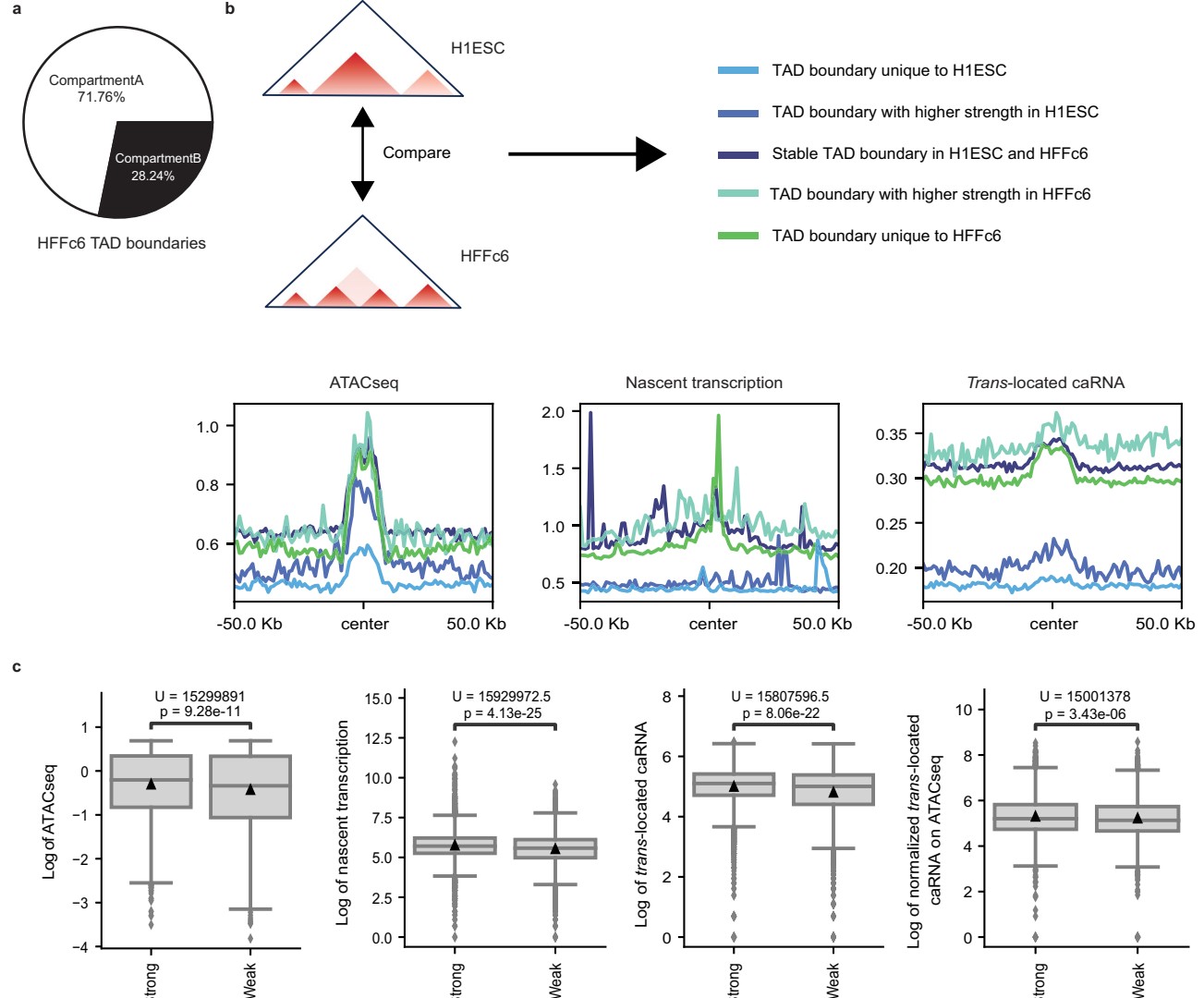

**Fig. 2 | *Trans*-located caRNAs are particularly enriched at TAD boundaries.**
**a** The percentage of HFFc6 TAD boundaries located in A and B compartments.
**b** Chromatin accessibility (ATAC-seq, left panel), and the abundance of nascent transcripts (middle panel) and *trans*-located caRNAs (right panel) at TAD boundaries (center) and their flanking regions in HFFc6. **c** Chromatin accessibility, the abundance of nascent transcripts and *trans*-located caRNAs, and the abundance of *trans*-located caRNAs normalized to chromatin accessibility, at strong (*n* = 7175) versus weak (*n* = 3971) TAD boundaries. The center line and triangle within the box represent the median and mean value, respectively. The box represents the interquartile range (IQR), with whiskers set to 1.5 times the IQR. Outliers are shown in points. Two-sided Mann-Whitney U tests were used to evaluate the differences between strong and weak TAD boundaries. U statistics and *p*-values are shown in the plot. Source data are provided as a Source Data file.

(ATAC-seq) (Fig. 2b). These results suggest the potential involvement of *trans*-located caRNAs at TAD boundaries and their contribution to TAD dynamics across cell types. Additionally, strong TAD boundaries exhibited significantly higher ATAC-seq and *trans*-located caRNA signals than did weak boundaries, and the association between boundary strength and *trans*-located caRNA abundance held after normalizing to the corresponding ATAC-seq signals (Fig. 2c). These results further indicate that the accumulation of *trans*-located caRNAs at TAD boundaries is not solely driven by DNA accessibility.

Unlike *trans*-located caRNAs that peaked at all HFFc6 TAD boundaries, nascent transcripts in HFFc6 mostly accumulated at TAD boundaries unique to HFFc6 (Fig. 2b). They also tended to be more frequent at strong TAD boundaries than weak ones (Fig. 2c). Overall, these results indicate that nascent transcripts could also contribute to the formation of TAD boundaries, particularly cell-type-specific ones, aligning with the enrichment of TAD boundaries at active promoters and the barrier function of RNAPs[3,13].

## CaRNAs increase the accuracy of 3D genome folding predictions
To learn how caRNAs contribute to 3D genome organization beyond TAD boundaries and in an unbiased way, we developed a deep learning framework called AkitaR. The models we implemented extend Akita[40] to predict chromatin interaction maps by incorporating both DNA sequence and RNA features extracted from nascent transcripts or *trans*-located RNAs (Supplementary Fig. 3). Similar to the original Akita, AkitaR uses 1D convolution neural networks to learn representations from ~1 Mb DNA sequence segments. The learned representations at the resolution of 2048 bp were subsequently concatenated with the RNA features, and dilated convolution neural networks were used to learn long-range dependencies. Lastly, 1D representations were averaged to 2D and further processed by dilated 2D convolutional neural networks to predict the ~1 Mb x 1 Mb contact matrices at 2048 bp resolution (Fig. 3a). These were contact frequencies after observed-over-expected normalization and log transformation (see Methods).

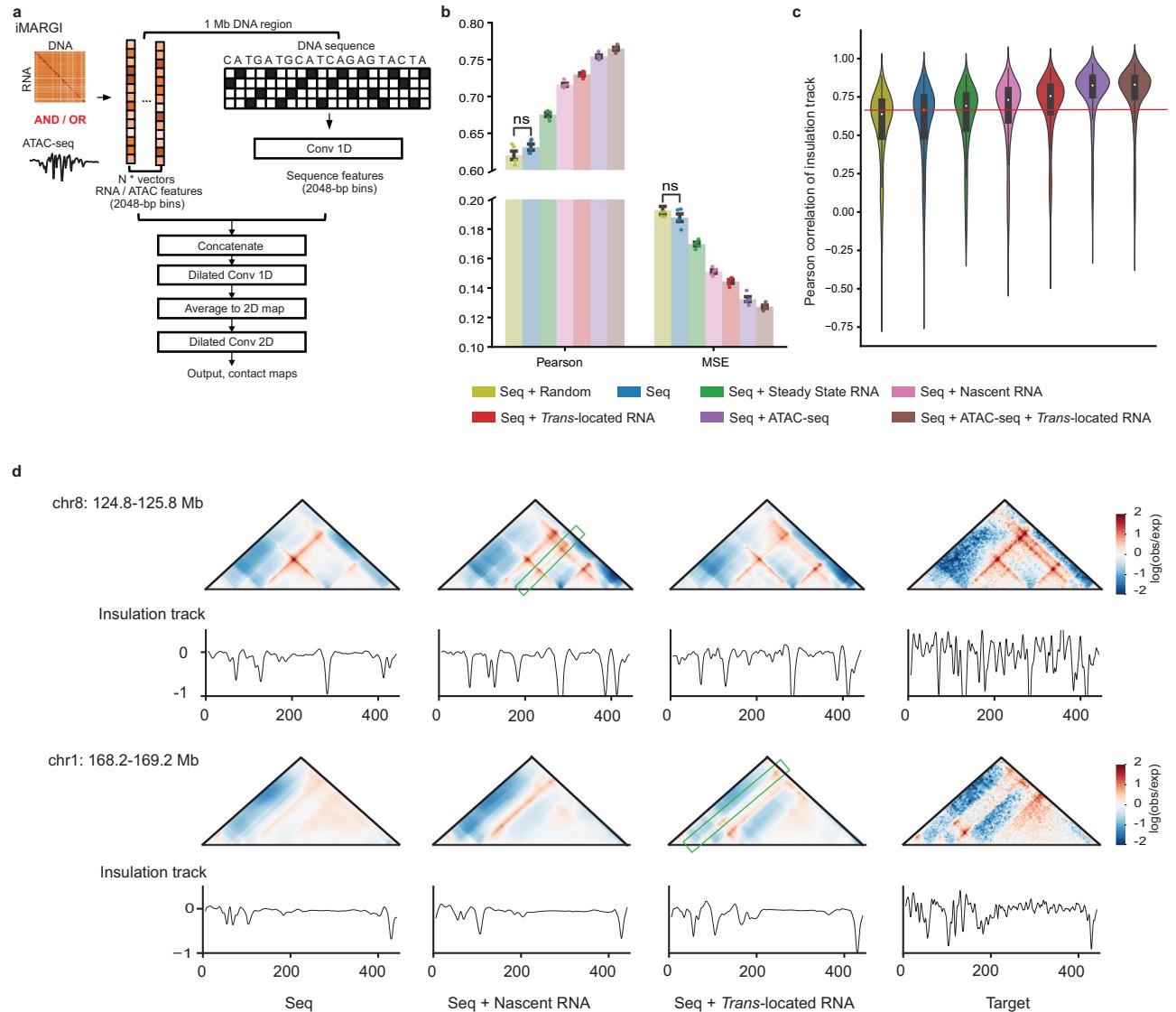

**Fig. 3 | Chromatin-associated RNAs contribute to accurate prediction of 3D genome folding. a** The architecture of the models in the AkitaR framework. **b** Barplots of performance for different types of AkitaR models on the held-out test set of genomic regions. Pearson's correlation and mean squared error (MSE) between experimental and predicted contact maps are used as performance metrics. Error bars represent the mean ± standard error of the mean for each model type independently trained five times. Two-sided Mann-Whitney U tests were used to evaluate differences between all pairs of models. Every comparison was significant (*p*-value < =0.05) except those labeled as not significant (ns). U statistics

and *p*-values for the comparisons are shown in Supplementary Data 1. Individual data points are shown as dots. **c** Violin plot of Pearson's correlation of insulation tracks from observed test set maps versus predicted maps (*n* = 413) for the best model of each type. The box represents the interquartile range (IQR), with whiskers set to 1.5 times the IQR. The dot within the box represents the median value. **d** Examples showing better prediction of contact maps with nascent transcripts (top panel) or *trans*-located RNAs (bottom panel). The 3D genome contacts with better prediction are highlighted with green rectangles. Source data are provided as a Source Data file.

We also designed additional models as controls or for comparison with the iMARGI based models (Supplementary Table 1 and Supplementary Fig. 3). For instance, since caRNAs are enriched in open chromatin, one of these models combined DNA sequence with features from chromatin accessibility (ATAC-seq) or ATAC-seq plus *trans*-located caRNAs. To disentangle the expression level of RNAs from their DNA contact frequencies and from nascent transcription, we incorporated steady-state transcription (RNA-seq). A control model with randomized signals from a standardized normal distribution was also built to alleviate the possibility that the improved performance was solely due to more features as input. Natural log transformations were applied on the RNA or open chromatin features before model fitting.

We found that all models with additional informative features achieved better predictions than the model with DNA sequence alone

as input (Fig. 3b–d, Supplementary Data 1 and Supplementary Figs. 4–6). This is consistent with results from models that incorporate epigenetic features such as CTCF binding or histone modifications[44]. Of the three RNA features, *trans*-located caRNA signals led to the AkitaR model with the highest performance, closely followed by nascent RNA, and then steady-state transcription (Fig. 3b, d, Supplementary Data 1, Supplementary Figs. 4, 6). On the other hand, at some regions, nascent RNA signals contributed to more accurate predictions than *trans*-located RNA inputs did (Fig. 3d, Supplementary Fig. 5). These results suggest that all the RNA features carry useful information about 3D genome folding, particularly *trans*-located caRNAs, though nascent transcription is more helpful at some loci. Adding chromatin accessibility signals yielded better performance than adding RNA features did (Fig. 3b, Supplementary Data 1 and Supplementary Fig. 4). However, adding *trans*-located caRNA plus chromatin accessibility

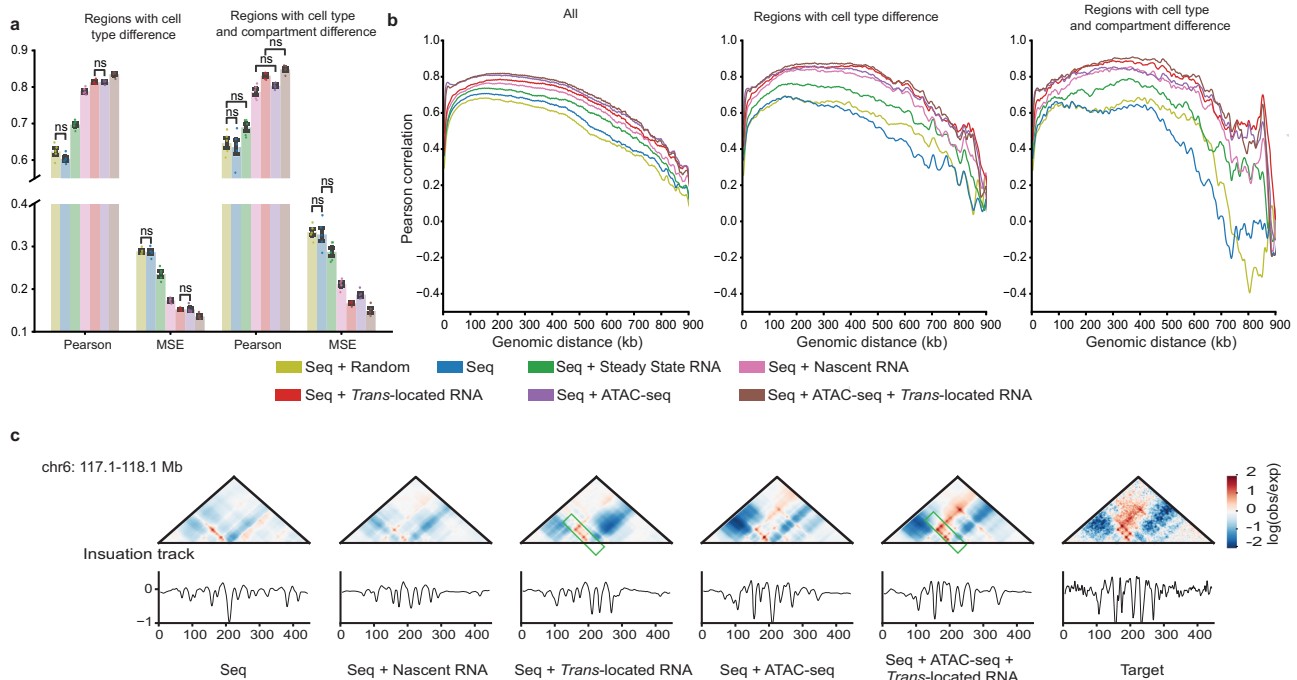

**Fig. 4 | Chromatin-associated RNAs help predict cell-type-specific genome folding. a** Barplots of Pearson's correlation and mean squared error (MSE) between experimental and predicted contact maps on the cell-type-specific subsets (MSE > 0.3) and cell-type-specific subsets (MSE > 0.3) with compartment changes from B compartment in H1ESC to A compartment in HFFc6. Error bars in the barplots represent the mean ± standard error of the mean for each model type independently trained five times. Two-sided Mann-Whitney U tests were used to evaluate differences between all pairs of models. Every comparison was significant (*p*-value < =0.05) except those labeled as not significant (ns). U statistics and *p*-values for the comparisons are shown in Supplementary Data 1. Individual data points are shown as dots. **b** Stratified Pearson's correlation between experimental and predicted contact maps for the best model of each type on the held-out test set, cell-type-specific subsets of test regions identified by MSE (MSE > 0.3) and cell-type-specific subsets (MSE > 0.3) with compartment changes from B compartment in H1ESC to A compartment in HFFc6. **c** An example showing the contribution of *trans*-located RNAs to the prediction of some chromatin interactions. The regions with better prediction are highlighted with green rectangles. Source data are provided as a Source Data file.

signals achieved even better performance than chromatin accessibility signals alone (Fig. 3b, Supplementary Data 1 and Supplementary Fig. 4), suggesting that RNA-DNA interactions provide additional information beyond marking open chromatin. In support of this hypothesis, we found that incorporating *trans*-located caRNAs into the models increased the correlation between predicted and observed insulation signals at TAD boundaries (Fig. 3c). Thus, deep learning clearly highlights the information that RNA-DNA interactions carry about chromatin organization.

## CaRNAs help predict cell-type-specific genome folding
Since RNAs, particularly ncRNAs, are often expressed in cell-type-specific ways[52], we hypothesized that the performance boost provided by incorporating RNA features into the AkitaR models would be most notable in regions with cell-type-specific genome folding. To evaluate this hypothesis, we first identified test regions that showed the largest differences in chromatin organization between H1ESC and HFFc6 based on MSE (34 regions) or MSE plus stratum-adjusted correlation coefficient (SCC) and structural similarity index measure (SSIM) (109 regions; see Methods) (Supplementary Fig. 7). We then evaluated the performance of our models in these cell-type-specific regions, and found that they showed a notably larger performance gap between models with additional features and the model with DNA sequence alone as compared to the ensemble of all test regions (Figs. 3b and 4a, b, Supplementary Data 1, Supplementary Figs. 8, 9). This finding demonstrates the capability of the AkitaR models to capture dynamic chromatin organization. Since genome compartmentalization correlates with RNA-chromatin interaction[20], we further evaluated model performance on cell-type-specific regions with compartment changes. Interestingly, the model with *trans*-located caRNA signals

achieved similar or even better performance than the model with chromatin accessibility signals on cell-type-specific regions with a compartment transition from B in H1ESC to the more active A compartment in HFFc6 (Fig. 4a, b, Supplementary Data 1, Supplementary Figs. 8, 9). This suggests the potential association of *trans*-located caRNAs with compartment transitions. Furthermore, by visually checking the cell-type-specific regions with better predictions from the models incorporating *trans*-located caRNAs, we observed that *trans*-located caRNAs helped capture some cell-type-specific chromatin interactions better than all other RNA and ATAC-seq features (Fig. 4c, Supplementary Fig. 10), prompting us to explore where these interactions mapped and what caRNAs they involved.

## Nuclear landmarks associate with *trans*-located caRNAs
To identify the caRNAs that contribute to 3D genome organization and the DNA regions to which they associate, we used DeepExplainer[53,54], which allowed us to quantify the importance of each RNA and ATAC-seq feature to the contact map predictions. DeepExplainer generates a score for each feature at each 2048 bp bin (see Methods). A negative score indicates that contact frequency decreases when the feature increases at that bin (e.g., caRNA is associated with loss of a chromatin loop or increased insulation at a TAD boundary); a positive score indicates that contact frequency rises when the feature increases. We observed that the distributions of contribution scores for the multiple RNA types were asymmetric, with slightly elongated left tails (Supplementary Fig. 11), hinting that RNA-DNA interactions may be more linked to insulation than enhancing chromatin interactions. Though the absolute contribution scores of caRNA features showed moderate positive correlation with caRNA signals, many genomic bins with high caRNA signals received low contribution scores, indicating that our

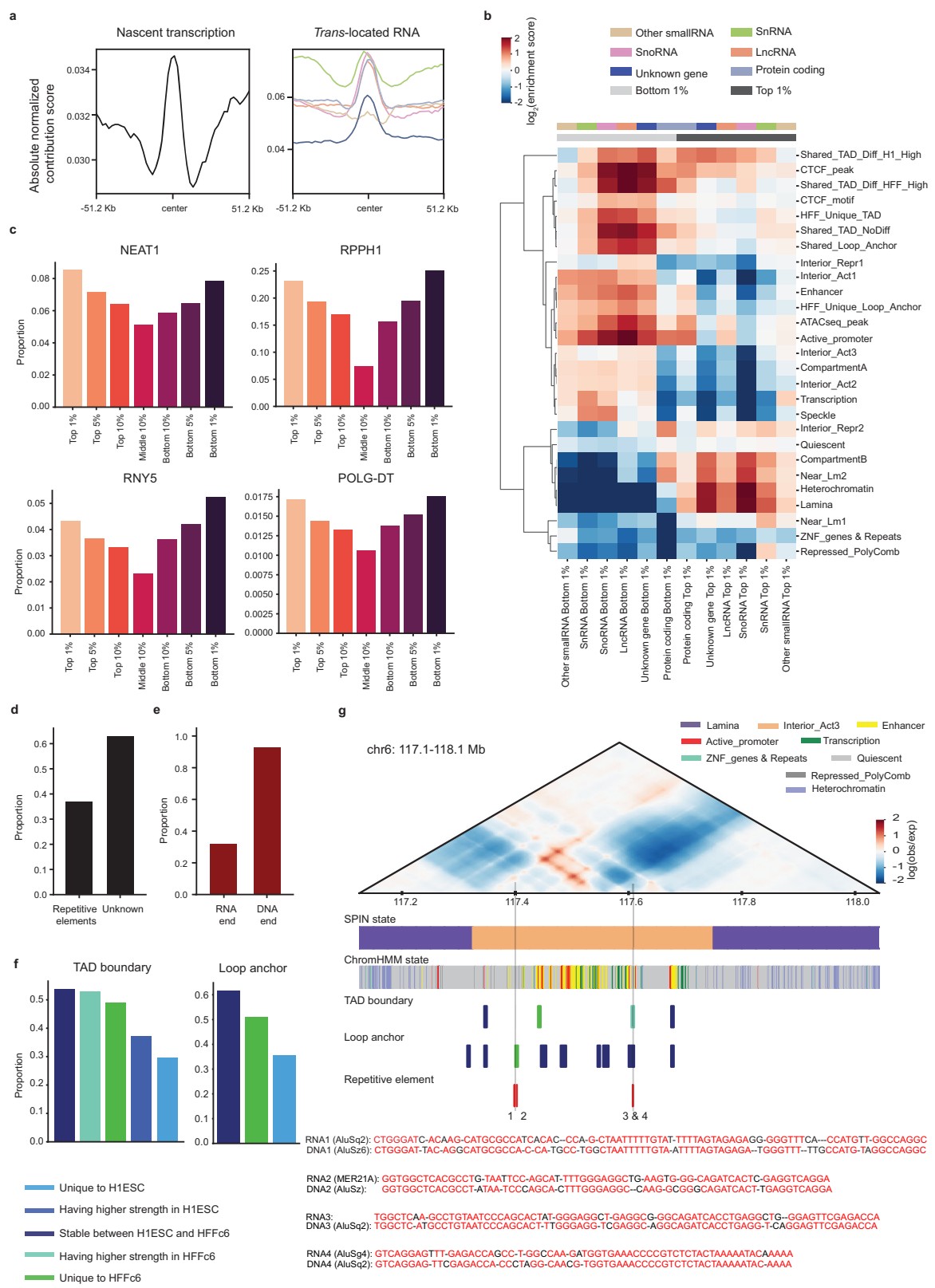

models learned where the caRNAs might contribute to genome folding (Supplementary Fig. 12). *Trans*-located caRNAs, which we already showed are enriched at TAD boundaries (Fig. 2b, c), tended to have higher absolute contribution scores at TAD boundaries than at their flanking regions (Fig. 5a). In contrast, the contribution scores of nascent transcripts were less elevated at TAD boundaries and remained

high in flanking regions, as we might expect when there is active transcription within TADs.

To test the hypothesis that AkitaR has learned a relationship between caRNA features and TAD boundary insulation, we performed two simulations. First, we generated 1000 random sequences of ~1 Mb and introduced TAD boundaries at randomly selected loci by inserting

**Fig. 5 | Chromatin-associated RNAs are associated with TAD boundaries, loop anchors and nuclear structures. a** The absolute contribution scores of nascent transcripts (left panel) and *trans*-located caRNAs (right panel) at TAD boundaries and their flanking regions. **b** Heatmap showing enrichment (log$_2$ enrichment score) of genomic regions with top 1% (positive) and bottom 1% (negative) contribution scores of each type of caRNA across TAD boundaries, loop anchors, Spatial Position Inference of the Nuclear genome (SPIN) and chromHMM states. **c** Example of four RNAs that preferentially interact with genomic regions with high absolute contribution scores (top 1%, top 5%, top 10%, bottom 1%, bottom 5%, bottom 10%) rather than with regions with lower absolute contribution scores (middle 10%). **d** The proportion of RNAs transcribed from repetitive elements for the interactions between DNA and RNAs derived from unknown genes. **e** The proportion of the RNA sequences in the candidate interactions that might form R-loops originated from Alu sequences and the proportion of the corresponding DNA sequences were annotated as Alu elements. **f** The proportion of TAD boundaries (left panel) and

loop anchors (right panel) having RNA-DNA interactions that could form R-loops. **g** An example locus showing candidate R-loops at Alu elements illustrating the contribution of *trans*-located RNAs to the prediction of chromatin interactions. The Alu elements that may form R-loops with RNAs at the loop anchors of a chromatin interaction that was specifically predicted by the model with *trans*-located caRNAs are shown in the track denoted Repetitive element. The candidate R-loops are numbered as 1, 2, 3 and 4 at the associated Alu elements. The best local alignment (with gaps) of the RNA and DNA sequences of the candidate R-loops (the complementary DNA sequences to RNAs are not shown) is shown. The nucleotides matching between RNA and DNA sequences are highlighted in red. Interior_Act 1: Interior Active 1, Interior_Act 2: Interior Active 2, Interior_Act 3: Interior Active 3, Interior_Repr1: Interior Repressive 1, Interior_Repr2: Interior Repressive 2, Near_Lm1: Near Lamina 1, Near_Lm2: Near Lamina 2. Source data are provided as a Source Data file.

convergent CTCF motifs. Progressively adding 1 to 4 CTCF motifs to increase the insulation strength led the sequence model to predict a decrease in average contact frequency (Supplementary Fig. 13a), suggesting a possible negative correlation between insulation strength and average contact frequency. Next, we selected 60 regions from the held-out test dataset with relatively simple structures and gradually increased the caRNA signal of each RNA type at selected TAD boundaries. We observed an increase of insulation strength for most of the RNA types, and the majority of them showed a decrease of the average contact frequency, including lncRNAs (Supplementary Fig. 13b). These simulation results suggest that AkitaR has inferred a potential causal role of *trans*-located caRNAs in strengthening TAD boundaries.

To further characterize the regions where caRNAs might shape chromatin organization, we ranked the contribution scores for each feature type and identified the genomic regions with scores in the top (positive contribution scores) or bottom (negative contribution scores) 1%. We found that the regions with bottom snRNA scores were preferentially located in nuclear speckles, aligning with their well-established roles in pre-mRNA splicing within nuclear speckles (Fig. 5b)[55,56]. Additionally, we observed that regions with top snoRNA scores were enriched in loci annotated as Interior_Repr2 (Interior Repressive 2) by SPIN (Fig. 5b), which was putatively associated with nucleoli[49] where snoRNAs function[57]. These two expected associations validate the capability of AkitaR to capture the functional roles of caRNAs.

Beyond these cases, we observed that genomic regions with bottom scores across RNA types were enriched at active chromatin, CTCF peaks, active promoters, enhancers, TAD boundaries and loop anchors (Fig. 5b). LncRNAs, snoRNAs and RNAs from unknown genes were particularly enriched at CTCF peaks, stable TAD boundaries between H1ESC and HFFc6, and shared TAD boundaries with higher insulation in HFFc6 (Fig. 5b), suggesting that these RNAs contribute to TAD boundaries, potentially by recruiting or stabilizing CTCF, in active chromatin. Interestingly, snoRNAs showed enrichment in nuclear speckles, consistent with the increasing evidence of the regulatory roles of some box C/D snoRNAs in alternative splicing[58–60]. On the other hand, regions with top scores were predominantly found in heterochromatin, particularly near lamina or at lamina associated regions (Fig. 5b), indicating that caRNAs play different roles at active and repressed chromatin, potentially via different mechanisms. CaRNAs from protein coding genes, however, showed very different patterns from all other caRNAs, with bottom rather than top scoring bins being enriched in compartment B and in particular at TAD boundaries with elevated insulation in HFFc6. Extending this analysis to the top and bottom 5% regions produced similar patterns, indicating that the enrichments are robust to the contribution score threshold (Supplementary Fig. 14).

In contrast to these nuanced patterns that differ across caRNA types and between active versus repressed chromatin, ATAC-seq features were significantly enriched in active chromatin, regardless of

whether they had top or bottom scores (Supplementary Fig. 15). These findings suggest that AkitaR captures differences between caRNA-DNA interactions and chromatin accessibility, motivating us to explore specific *trans*-located caRNAs. To further disentangle the independent effects of caRNAs beyond their association with open chromatin, we identified specific DNA regions where *trans*-located caRNAs have high absolute contribution scores and ATAC-seq features do not (absolute normalized contribution score > 0.25, |fold change| >5) (Supplementary Fig. 16). These regions showed similar chromatin and SPIN state enrichments as the top and bottom scoring regions more generally (Supplementary Fig. 16), confirming the contribution of *trans*-located caRNAs to chromatin features beyond chromatin accessibility.

## CaRNAs may form *trans* R-loops with Alu sequences

To identify the caRNAs with the largest contributions to AkitaR's chromatin map predictions, we ranked them based on their association with DNA regions that have high absolute contribution scores (top or bottom 5% for each RNA type; Supplementary Data 2). We observed that the top 10 RNAs were all highly prevalent in HFFc6 (Supplementary Fig. 17 and Supplementary Data 2). These included multiple ncRNAs previously known to play roles in chromatin structures, such as lncRNAs *MALAT1* and *NEAT1* and snRNAs *RNU2-2P*, *RNU12* and *RN7SK*[16,26,29,35,61,62] (Fig. 5c, Supplementary Fig. 18 and Supplementary Data 2). Interestingly, all top 10 snoRNAs are C/D box snoRNAs, including *SNORD47*, *SNORD79* and *SNORD27* (Supplementary Fig. 18). Beyond these, many RNAs without previous evidence for roles in chromatin organization stood out, such as lncRNAs *RNY5*, *RPPH1*, *POLG-DT* and differentially expressed lncRNAs between H1ESC and HFFc6, such as *THBS1-IT1* and *ENSG00000260772*. In addition, these lncRNAs were preferentially associated with regions with high absolute contribution scores compared to the regions with low scores (Fig. 5c, Supplementary Fig. 18). Since the pattern of enrichment of these caRNAs mirrors that of *MALAT1* and *NEAT1*, we hypothesize that these caRNAs also play mechanistic roles in 3D genome organization.

To explore the caRNAs that might shape genome structure over chromatin accessibility, we investigated the caRNAs (top10) that were preferentially associated with genomic regions having higher absolute *trans*-located caRNA contribution compared to chromatin accessibility. We identified a list of RNAs that was nearly identical to the one identified genome-wide (Supplementary Data 2). However, some RNAs were found to preferentially interact with these differentiated regions but were not enriched in top or bottom scoring regions overall. These included the ncRNAs *ZNRF3-AS1*, *MIR6726* and *MIR4796*, which not only showed higher interaction ratios with the differentiated regions but also interacted with more than one of these DNA regions (Supplementary Data 3 and Supplementary Fig. 19). These caRNAs are high-confidence candidates for contributing to nuclear structures in specific ways beyond being generally associated with accessible chromatin.

Since genomic regions with bottom contribution scores from RNAs of unknown genes were enriched at TAD boundaries and loop anchors, we further explored the interactions between DNA and RNAs derived from unknown genes. We found that around 37% of these RNAs were transcribed from repetitive elements (Fig. 5d). Since Alu sequences were proposed to promote long-range enhancer-promoter interactions, possibly through R-loops[25,63], we aligned the sequences of each pair of DNA-RNA *trans* interactions in HFFc6 using the local alignment function of the python module pairwise2 in search of potential candidates for R-loop formation. Around 0.3% of *trans* interactions were considered as candidates by exhibiting over 80% identity between RNA and DNA sequences plus continuous, uninterrupted perfect matches exceeding 10 base pairs. We found that 32% of the RNA sequences in these candidate interactions originated from Alu sequences, and 93% of the DNA sequences were annotated as Alu elements (Fig. 5e). Furthermore, these candidate interactions tended to increase at stable TAD boundaries, TAD boundaries having higher insulation strength in HFFc6 or unique to HFFc6 in contrast to TAD boundaries with higher strength in H1ESC or unique to H1ESC (Fig. 5f). The same trend was also observed for loop anchors (Fig. 5f), aligning with the roles of Alu sequences in long-range enhancer-promoter interactions. More importantly, both loop anchors of the cell-type-specific interaction that was captured by the models with *trans*-located caRNAs but no other models in Fig. 4c could form *trans* R-loops at Alu sequence loci (Fig. 5g). This provides a potential mechanism for loop formation at loci with *trans* Alu RNA-DNA interactions, demonstrating the capability of the AkitaR model to capture these interactions and generate testable, mechanistic hypotheses.

## Discussion

In this study, we proposed deep learning models that leverage both DNA sequence and the distribution of caRNAs across the genome to predict chromatin interaction maps. Both nascent transcripts and *trans*-located RNAs contributed to these AkitaR models being able to make more accurate predictions than with sequence alone, especially in regions of the genome with different folding between cell types. While the magnitude of improvement was modest, it was statistically significant, and the models also learned the importance of caRNAs at chromatin features, such as CTCF peaks, TAD boundaries and loop anchors. Amongst the caRNAs that preferentially interacted with genomic regions having high absolute *trans*-located caRNA contribution scores were RNAs with known roles in chromatin structure, such as snRNAs at regions located in nuclear speckles, snoRNAs in nucleoli, and the lncRNAs *MALAT1* and *NEAT1*. These validations gave us confidence in our discovery of several RNAs that might be involved in the regulation of chromatin organization in HFFc6, including *RNY5*, *RPPH1*, *POLG-DT* and *THBS1-IT1*.

Since *trans*-located RNAs tended to be enriched at open chromatin regions and the model with chromatin accessibility signals achieved better performance than the one with *trans*-located RNA signals, it might be argued that *trans*-located RNAs diffused randomly and that their enrichment in these regions solely reflected the accessibility of chromatin. However, our results suggest that *trans*-located RNAs may play causal roles in genome folding on top of randomly diffusing to distant DNA regions. Firstly, *trans*-located caRNA signals were found to be higher at strong TAD boundaries compared to weak ones even after being normalized on ATAC-seq signals. Secondly, stable TAD boundaries, TAD boundaries with higher insulation strength in HFFc6 or the ones unique to HFFc6 also tended to have RNA-DNA interactions that might form *trans*-acting R-loops. Moreover, the model with both chromatin accessibility and *trans*-located caRNAs also slightly outperformed the model with only the chromatin accessibility. Additionally, the model with *trans*-located caRNA signals achieved better performance on some subsets of the test regions

compared to the one with chromatin accessibility signals. Particularly, we observed that some local chromatin features were only accurately predicted by the models with *trans*-located caRNA signals as input. Their loop anchors might be mediated by *trans*-acting R-loops formed at Alu sequences. Lastly, genomic regions with high absolute contribution scores from different RNA types also showed enrichment in chromatin and SPIN states both in active and repressed chromatin in contrast to the enrichment of genomic regions with high absolute ATAC-seq contribution scores only in active chromatin. On the other hand, the chromatin being open could also have been the result of the binding of *trans*-located caRNAs. For example, lncRNA *ROR* is reported to promote H3K4me3 at *TIMP3* genes in *trans* by recruiting MLL1[64], a histone methyltransferase that trimethylates H3K4[65]. Additionally, R-loops, which may form in *trans*, have been observed to extensively overlap with H3K4me3 peaks[66] and reduce DNA methylation at promoter regions[67,68]. Moreover, RNAs can open up the chromatin in a rapid manner by neutralizing positively charged histone tails with their negative charges[69]. Chromatin-associated RNAs could also help maintain an open chromatin structure by binding with structural chromatin-associated proteins, such as SAF-A[70] and Df31[71], and the depletion of RNAs leads to chromatin compaction[69,71–73]. Together these results indicate that AkitaR is useful not just as a predictive model but also as an explanatory method for generating testable hypotheses about the functions of caRNAs.

Although most of the eight RNA types increased the accuracy of the chromatin map predictions, each type was associated with somewhat distinct chromatin structures. For example, snoRNAs, lncRNAs and RNAs from unknown genes showed more enrichment at CTCF peaks, shared TAD boundaries and loop anchors than did snRNAs, whereas lncRNAs were more enriched at promoters and enhancers compared to other RNA types. Moreover, besides snRNAs, we observed the enrichment of snoRNAs in nuclear speckles. While it is well-established that snoRNAs have vital functions within the nucleolus, growing evidence suggests that some snoRNAs, including *SNORD27*, *SNORD88C*, and *SNORD115*, may exert regulatory influence over the alternative splicing of pre-mRNAs that originate from distantly located genomic loci[58–60]. Finally, we noticed that regions with positive contribution from RNAs of different types, particularly RNAs from unknown genes, showed enrichment at heterochromatin and lamina or near lamina regions. This is consistent with the recent evidence that ncRNAs, especially repetitive ncRNAs, play roles in anchoring specific genomic loci to nuclear lamina or recruiting H3K9me3-related methyltransferases to promote heterochromatin[23,74,75].

The high performance of our AkitaR models allowed us to explore the contribution of caRNAs in genome organization in an unbiased and effective way. Leveraging feature importance scores or high-throughput in silico screening, we could efficiently prioritize candidate genomic loci that are dependent on caRNAs for accurate genome folding and develop hypotheses for functional characterization with additional analyses. These hypotheses could be further validated with experimental techniques, such as genome engineering, RNA inhibition or RNA overexpression, in the context of 3D genome folding. We anticipate that this strategy of integrating deep learning models with bioinformatics analyses will drive the generation of hypotheses and accelerate wet lab discoveries.

While AkitaR offers us an effective way to unravel the roles of *trans*-located caRNAs in genome folding, our approach has several limitations. First, the genome-wide RNA-chromatin interaction data that we used to extract the *trans*-located RNA features were limited to several cell types, making it difficult to generalize our models and analyses to a wide range of cellular contexts. Secondly, as *trans*-located caRNA signals might somewhat reflect the accessibility of chromatin, models may face challenges in distinguishing which regions *trans*-located caRNAs play a driver role and which regions they act as

passengers. Lastly, many of the RNAs might only function in *trans* at limited regions, and our analyses based on genome-wide signals might not be able to capture the contribution of these RNAs.

In summary, we investigated the roles of caRNAs, particularly *trans*-located caRNAs, in regulating 3D genome folding by genome-wide analyses and deep learning models. We showed the contribution of both nascent transcripts and *trans*-located caRNAs to genome organization. These analyses provide insights and generate testable hypotheses about the roles of caRNAs in chromatin organization.

## Methods

### Micro-C data and processing
High-quality Micro-C datasets mapped to hg38 in .pairs format for HFFc6 and H1ESC were downloaded from the 4DN data portal (https://data.4dnucleome.org/)[46,47] and processed into 2048-bp ($2^{11}$ bp) bins, followed by genome-wide iterative correction (ICE) normalization, adaptive coarse-graining, observed over expected normalization, log transformation, clipping to (−2,2), linear interpolation and convolution with a 2D Gaussian filter for smoothing[40]. These data and their paired genome were further divided into training (7009), validation (419) and test (413) examples, each of which was a ~1 Mb ($2^{20}$ bp) region.

Annotations of compartment and TAD boundaries for the Micro-C datasets identified by cooltools at the resolution of 25 Kb and 5 Kb, respectively, were also downloaded from the 4DN data portal[76]. TAD boundaries with insulation strength between 0.2 and 0.5 were considered as weak boundaries and the ones with strength larger than 0.5 were defined as strong boundaries. The TAD boundaries in a cell type that were within 20 Kb of the TAD boundaries from the other cell type were defined as shared TAD boundaries, otherwise they were considered as cell type unique TAD boundaries. The $\log_2$ fold change of insulation strength for the shared TAD boundaries were further calculated and used to classify them into stable TAD boundaries between H1ESC and HFFc6 with no insulation difference ($|\log_2(HFFc6/H1ESC)| <= 1$), shared TAD boundaries with higher insulation strength in H1ESC ($\log_2(HFFc6/H1ESC) < -1$) and shared TAD boundaries with higher insulation strength in HFFc6 ($\log_2(HFFc6/H1ESC) > 1$).

Chromatin loops at 5 Kb and 10 Kb resolution for the Micro-C datasets were identified using HiCCUPS[7]. Similar to TAD boundaries, the loop anchors were classified as shared loop anchors and cell type unique loop anchors with distance limit of 20 Kb.

### iMARGI data and processing
iMARGI data in .pairs format for HFFc6 and H1ESC on hg38 were obtained from the 4DN data portal and converted into contact matrices at the resolution of 10-bp (for preliminary analyses), 2048-bp (for model inputs), and 5000-bp (for analyses at TAD boundaries and loop anchors) after removing low-quality mappings (MAPQ $<= 30$)[20]. Nascent transcription was estimated as the number of reads with their RNA ends mapped to each bin (10-bp/2048-bp/5000-bp) in the contact matrices (log value for model input). In order to get the signals of *trans*-located RNAs at each bin for the *trans*-located caRNA model, the interactions between RNAs and DNAs within ~1 Mb ($2^{20}$ bp) linear distances were filtered out. The self-interactions between genes that are longer than ~1 Mb ($2^{20}$ bp) were also removed. Considering the potentially distinct roles of different RNA types, we annotated the RNA ends of the contact matrices with comprehensive genes from GENCODE (v43). We noticed that many snoRNA genes annotated in Refseq were missed in GENCODE but showed high expression in iMARGI data. We thus incorporated the annotations of snoRNAs from Refseq into GENCODE. We then classified the bins in the RNA end into eight groups based on their overlap with the transcription sites of different types of RNAs, which are snRNAs, snoRNAs, other small RNAs, lncRNAs, miscellaneous RNAs, RNAs from protein-coding genes, RNAs from all other types of genes and RNAs from regions without known gene annotations. The total number of reads from all RNAs in each RNA group with their DNA ends mapped to a bin was calculated as the *trans*-located caRNA signal of that bin from the RNA group. Log transformation was performed for model input.

Besides gene annotations, the RNA and DNA end of the iMARGI interactions were annotated for repetitive elements with data from the RepeatMasker database for downstream analyses[77].

### RNA-seq and ATAC-seq data
RNA-seq and ATAC-seq data in .bigWig format for HFFc6 were downloaded from the 4DN data portal, respectively[47,78,79]. Log values of the normalized signals of each 2048-bp bin on the library size of iMARGI data were extracted from the data to get the input for the model with steady-state transcription level or the model with chromatin accessibility. The signals of ATAC-seq at 5000-bp bins were also calculated for the analyses at TAD level.

### Model architecture, training and evaluation
AkitaR was extended from Akita to predict 3D genome folding by using both DNA sequence and RNA / ATAC-seq signals. We kept the head of Akita and adjusted the trunk architecture by concatenating the above RNA / ATAC-seq features of length 512 to the vector representations of DNA sequence. The DNA representation was the output of 11 convolution blocks, each of which included convolution, batch normalization and max-pooling layers. Keeping hyperparameters the same as Akita, the model was trained to maximize Pearson's correlation coefficient between experimental maps and predictions. We chose to optimize on Pearson's correlation coefficient over mean squared error (MSE) because pixel-wise MSE tends to be very sensitive to noise[80]. To ensure robust comparison between different models, each model was trained five times, and the one with the best performance based on both MSE and Pearson's correlation was selected for downstream analyses and contact map visualization.

Model performance was evaluated on the test dataset using MSE, SCC, Pearson's and Spearman's correlations. Correlations were calculated using the functions in the python SciPy library. To examine the capability of the models to capture cell-type-specific regions, two subsets of test regions that had different contact maps between H1ESC and HFFc6 were selected based on MSE, SCC and SSIM. One cell-type-specific subset included the regions with high MSE ($>0.3$) between H1ESC and HFFc6 experimental maps. The other one consisted of not only regions with high MSE ($>0.3$), but also those with low SCC ($<0.2$) or SSIM ($<0.08$). Here, SCC is the weighted sum of Pearson's correlation for each stratum and shares the similar range as Pearson's correlation coefficients[81]. Since both the predicted and experimental contact map used in this study were normalized against distance dependent decay, SCC is highly consistent with Pearson's correlation. SSIM is a widely used metric in imaging studies that qualifies the similarity between two images[82]. To further evaluate whether RNAs were associated with the compartment changes between cell types, the cell-type-specific subsets were further divided into the ones without compartment change, the ones that switched to compartment B in HFFc6 from compartment A in H1ESC and also the ones that changed to compartment A in HFFc6 from compartment B in H1ESC.

### Insulation scores
Insulation profiles of experimental and predicted contact maps were identified by sliding along each diagonal bin of the contact matrix using a diamond-shaped window and calculating the average contact frequency within the window[83]. The bins at the end of the diagonal were ignored for calculation.

### *Trans*-located proportion of RNAs
iMARGI data in .pairs format was first converted into .bedpe format. The total number of reads with RNA end mapped to each gene was

calculated as its nascent transcription. Then the read pairs with their DNA and RNA end within ~1 Mb ($2^{20}$ bp) linear distances were removed and the resulting reads mapped to each gene was regarded as its *trans*-located abundance. The ratio of the *trans*-located abundance to its nascent transcription was calculated as the *trans*-located proportion of each RNA gene. To better distinguish the roles of host genes and the genes within them, the reads mapped to the genes within them were subtracted from the host genes.

### Signals at TAD boundaries and flanking regions
Nascent transcription and *trans*-located caRNA signals at the resolution of 10-bp were first converted into bigWig format using bedGraphToBigWig[84]. Then the signals of ATAC-seq, nascent transcription and *trans*-located caRNAs at TAD boundaries and their flanking regions were calculated using deepTools computeMatrix from the bigWig files and plotted using deepTools plotHeatmap[85].

### CTCF ChIP-seq and binding sites
CTCF ChIP-seq peaks and signals were downloaded from ENCODE data portal[86]. The genome-wide CTCF sites identified by FIMO using all three CTCF PWMs in JASPAR database with *p*-value less than 1e-5 were downloaded from the R resources AnnotationHub[87].

### ChromHMM state and SPIN state
ChromHMM were employed to annotate the chromatin regions of H1ESC and HFFc6 using six epigenomes (H3K27ac, H3K4me1, H3K4me3, H3K9me3, H3K27me3 and H3K36me3), which were downloaded from the ENCODE data portal[86]. We defined 18 chromatin states using the default parameters and annotated them by checking their enrichment for gene related regions (transcription start site (TSS), transcription end site, gene body, exon and intron), repetitive elements and epigenetic peaks. The ones annotated as active TSSs, TSS flanking regions or bivalent promoters were extracted as active promoters and the ones annotated as active enhancers, genic enhancers, weak enhancers and bivalent enhancers were obtained as enhancer regions and used for downstream analyses. Annotations of SPIN states[49] were obtained from Jian Ma's lab (Zhang et al. in preparation).

### Contribution scores
DeepExplainer (DeepSHAP implementation of DeepLIFT)[53,54] was employed to compute the contribution scores of the RNA and ATAC-seq features. For the examples in the validation and test dataset, randomly selected 20 examples from the training dataset were used as background. For the training dataset, we divided it into two subsets. For the first half, randomly selected examples from the second half acted as background, and vice versa. The contribution scores for each feature were normalized by dividing into their maximum absolute values.

The genomic regions with their contribution scores located within the top (positive) and bottom (negative) 1% and 5% for each feature were extracted for enrichment analyses. Specifically, their enrichment at CTCF sites, active promoters, enhancers, other chromHMM states, TAD boundaries, loop anchors and SPIN states were measured by calculating the ratio of observed over expected signals. To avoid the bias caused by the bins with positive signals, only the ones with positive input values were used in the enrichment analyses.

### Differential analyses of RNA contribution scores
The normalized scores of *trans*-located signals of each RNA type at each DNA bin were compared to the scores of ATAC-seq signals. The DNA bins with fold change greater than 5 and the absolute value of normalized contribution score larger than 0.25 were considered as the ones with differential contributions between *trans*-located caRNAs and ATAC-seq signals.

### Candidate RNA identification
A hypergeometric test was employed to evaluate whether RNA-DNA interactions occur more often than expected by random chance. The test assumes that each DNA bin has an equal probability to interact with any RNA in a random manner and each interaction is independent. The interactions with FDR < = 0.05 were extracted as high-confidence interactions. These high-confidence interactions were then used to identify RNAs that preferentially interacted with selected DNA bins.

### Simulations of increasing TAD insulations
Two different simulations were performed to explore how altering DNA sequence or caRNA signals is predicted to alter local insulation and chromatin contacts. First, one thousand random DNA sequences of $2^{20}$ bp were first generated using the SimDNA python package (https://github.com/kundajelab/simdna). TAD structures were then introduced by symmetrically inserting forward and reverse CTCF motifs at randomly selected loci between 0.15 and 0.85 of each DNA sequence. Following that, one to four convergent CTCF motifs were progressively added to TAD boundaries with distance from previously inserted CTCF motifs at 500 bp and the contact maps of the DNA sequences were generated using the sequence alone model. Second, 60 regions with relatively simple structures were selected from the test set, and the caRNA signals at selected TAD boundaries (1–2 for each region) were increased by folds of $e^{0.5}$, $e^1$, $e^{1.5}$, $e^2$, separately for each RNA type. Predictions were made with the model incorporating *trans*-located caRNA signals, and the resulting map predictions were compared to the starting map without elevated caRNA signals.

### Pairwise alignment of DNA and RNA sequences
To search for potential candidates of R-loop formation, we extracted the sequences of each pair of DNA-RNA *trans* interactions in HFFc6 and then aligned them using the pairwise2 sequence alignment module in the Bioptyhon package (local alignment)[88]. The ones with over 80% of RNA sequence matching to DNA sequences and continuous perfect matches exceeding 10 bp were considered as candidates.

### Statistical analysis
Two-sided Mann-Whitney U tests were used to compare strong versus weak TAD boundaries, model performance statistics and simulation scenarios with different numbers of inserted sequences or different levels of RNA features. Hypergeometric tests were employed to identify high-confidence, statistically significant RNA-DNA interactions.

## Data availability
Publicly available data used in this study can be found at: 4D Nucleome Data Portal (https://data.4dnucleome.org/) with accession numbers (1) Micro-C for HFFc6 and H1ESC: 4DNESWST3UBH, 4DNES21D8SP8, (2) iMARGI for HFFc6 and H1ESC: 4DNES9Y1GHK4, 4DNESNOJ7HY7, (3) ATAC-seq for HFFc6: 4DNESMBA9T3L, (4) RNA-seq for HFFc6: 4DNESFH3EHTU [https://data.4dnucleome.org/higlass-view-configs/d462e61d-88e5-48bc-969c-4c208412fea2/]; ENCODE data portal (www.encodeproject.org/) with accession numbers (1) ChIP-seq data, H3K27ac for HFFc6 and H1ESC: ENCSR510VXV, ENCSR880SUY, (2) H3K4me1 for HFFc6 and H1ESC: ENCSR340XKM, ENCSR000ANA, (3) H3K4me3 for HFFc6 and H1ESC: ENCSR639PCR, ENCSR000AMG, (4) H3K9me3 for HFFc6 and H1ESC: ENCSR938NXC, ENCSR000APZ, (5) H3K27me3 for HFFc6 and H1ESC: ENCSR129TUY, ENCSR186OBR, (6) H3K36me3 for HFFc6 and H1ESC: ENCSR519CMW, ENCSR000ANB, CTCF for HFFc6: ENCSR163ULN; R resources AnnotationHub for CTCF binding sites (https://github.com/mdozmorov/CTCF); GENCODE, NCBI (https://www.ncbi.nlm.nih.gov/datasets/genome/GCF_000001405.40/, RefSeq); UCSC (http://hgdownload.soe.ucsc.edu/goldenPath/hg38/database/, RepeatMasker). Pretrained models, model input,

contribution scores of RNA / ATAC-seq features as well as target map and test set predictions are available at Zenodo (https://zenodo.org/records/10015009). Other data used to generate the figures are available in the CaRNAs_in_Chromatin_Architecture github repository (https://github.com/shuzhenkuang/CaRNAs_in_Chromatin_Architecture). Source data are provided with this paper.

## Code availability

The code of AkitaR[89] (modified from Akita), custom code for data exploration and downstream analyses and a jupyter notebook for figure generation are available at https://github.com/shuzhenkuang/CaRNAs_in_Chromatin_Architecture.

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

## Acknowledgements

We gratefully acknowledge members of the Pollard lab for project feedback, Jian Ma and Yang Zhang for providing the SPIN state data, and Sheng Zhong's group for project suggestions and providing the iMARGI data. This work was supported by the NIH 4D Nucleome Project (grant #U01HL157989 to K.S.P.) and Gladstone Institutes. K.S.P. is an investigator of the Chan Zuckerberg Biohub San Francisco.

## Author contributions

S.K. and K.S.P. conceived and designed the work. S.K. conducted all analyses. S.K. and K.S.P. wrote the manuscript.

## Competing interests

The authors declare no competing interests.
