## [Peer Review File · Nature Communications]

Exploring the Roles of RNAs in Chromatin Architecture Using Deep LearningREVIEWER COMMENTS

Reviewer #1 (Remarks to the Author):

The chromatin is richly decorated with many chromatin-associated RNAs (caRNAs), begging the question of whether caRNA has widespread impacts on 3D genome organization. Kuang and Pollard developed a deep learning model AkitaR that jointly uses genomic sequence and the caRNAs attached to every genomic region to predict the 3D genome organization. Using AkitaR, Kuang and Pollard demonstrated that incorporating genomewide caRNA-DNA contact data in the deep learning model led to a more accurate recapitulation of 3D genome organization than using DNA sequence alone, supporting the idea that caRNA has widespread impacts on 3D genome organization. Furthermore, Kuang and Pollard demonstrated that trans-caRNAs, the caRNAs attached to the target genomic regions far away from the caRNA's transcription site, are particularly informative in recapitulating the 3D genome organization. This result holds even when the deep learning model has already taken chromatin accessibility information into account, revealing that caRNA-DNA contacts contain additional information about 3D chromatin features beyond chromatin accessibility. Based on AkitaR and in situ Mapping of RNA-Genome Interactome (iMARGI)-derived RNA-DNA contacts, Kuang and Pollard identify candidate R-loops formed by trans-caRNA and target genomic sequences, which frequently appear at loop anchors and stable TAD boundaries and are enriched with Alu transposons, alluding to a model where trans-caRNA-mediated R-loops promote chromatin looping and TAD boundary formation. These results demonstrate AkitaR's ability to generate hypotheses with deep mechanistic insights.

I recommend a speedy publication of this important work.

Minor comments:

1. The authors have cited two highly relevant iMARGI papers. Perhaps mentioning MARGI (Sridhar et al. Current Biology 2017 PMID: 28132817) in the Introduction and citing the original iMARGI paper (Yan et al. PNAS 2019 PMID: 30718424) could enrich the background information.
2. Line 350: should "not other models in Fig. 4C could form ..." read "no other models in Fig.

4C could form”?

Reviewer #2 (Remarks to the Author):

Exploring the Roles of RNAs in Chromatin Architecture Using Deep Learning

Corresponding author: K S Pollard

This paper explores the role of chromatin-associated RNA (caRNA) in the spatial chromatin architecture. In brief, it shows that caRNAs likely play a causal role in establishing tissue-specific chromatin architecture, and identifies a substantial number of novel RNA genes that are likely involved in this process.

Previously it was shown, by the authors and others, that chromatin contact maps can be predicted with reasonable accuracy from DNA sequence alone. Here the authors show, first, that caRNAs are associated with intranuclear spatial position, open chromatin, and TAD boundaries, and second that in a deep learning model (AkitaR), caRNAs improve the prediction of chromatin contact maps when used as input, compared to models that use only DNA sequence, or a combination of DNA sequence and chromatin accessibility data.

The authors argue that this is likely because caRNA plays a causal role in chromatin architecture. Although the improvement in predictive ability is modest, in comparison with a model that uses both DNA sequence and chromatin accessibility data, the authors reasonably suggest (l. 384-5) that this might be the result of the chromatin accessibility signal partly explaining away a causal signal from caRNA, which could happen if caRNA interaction result in chromatin becoming accessible. In favor of this explanation, the authors show that the caRNA signal is highly specific to certain genes; in particular, the 10 genes that contributed most strongly to the model's predictive ability include 5 with known roles in chromatin structure.

I also liked the in-silico experiment, of taking random genomic loci and adding convergent CTCF motifs which are known to be associated with isolation domains and reduced contact

frequency, and trans caRNAs of which the model's importance scores indicated an association with reduced contact frequency. The model indeed subsequently predicted reduced contact frequency for both types of synthetic sequence, suggesting a causal role. This is precisely the kind of in-silico experiments that are enabled by black-box machine learning models trained on extensive and complex data sets. Although the results of these experiments are never fully conclusive as they are based on observational data and confounding remains possible, nevertheless it is a good test, as it is easy to interpret, and the results are what you would expect if the causal relationship indeed exists.

Specific comments:

1. The observation that the model that uses DNA sequence and caRNA as input performs best (Fig 3b) is key to the paper. Please therefore add estimates of standard errors in the mean (or median) to the figure to indicate how precise these performance measures are. This probably involves re-training the model several times.

2. I believe the argument referred to in the 2nd paragraph above, and mentioned in l. 384-5, should be expanded a bit in the discussion, as I think this is a crucial point. Even if adding caRNAs to the model does not significantly improve contact map predictions, the fact that this model uses known RNAs to make those predictions, and the likely possibility that caRNA interactions result in accessible DNA, make the model useful as an explanatory rather than just a predictive model. The authors do argue this point, but only briefly (l. 384-5); I would suggest they expand on this part a little.

3. Selecting trans caRNAs with high importance scores makes the in-silico experiment (l. 259-69) is somewhat circular. I suggest to re-do this experiment with RNA genes that were previously known to be involved with regulation of chromatin structure as well as expressed in HFFc6, including the 5 genes mentioned in l. 312.

4. Thank you for the detailed data access information, and for making the code for exploratory data analysis and figures available. For full replicability and for a detailed

understanding of the model, the modified AkitaR code is also required. Please also make this available in the GitHub repository.

5. Fig 5b, Supp Figure 14: please use the same order in the legend as in the plot. Please state whether the numbers in the heat map refer to $2\log$, $10\log$ or natural log.

6. Supplementary Table 2, typo in the title.

7. l. 339 - pairwise2, a typo? Do you mean mutual best hit?

Response to reviewers' comments:

We would like to thank the reviewers for their valuable comments and helpful suggestions. We have addressed reviewers' comments below with our responses in **blue font** and made changes in the manuscript. To follow the formatting instructions and policy requirements, we have also made changes in all Figures to put the text in sentence case and implement other requested changes to the text and Figure legends. All changes in the main or supplemental text are highlighted in red.

Response to Review 1:

Comment 1: The chromatin is richly decorated with many chromatin-associated RNAs (caRNAs), begging the question of whether caRNA has widespread impacts on 3D genome organization. Kuang and Pollard developed a deep learning model AkitaR that jointly uses genomic sequence and the caRNAs attached to every genomic region to predict the 3D genome organization. Using AkitaR, Kuang and Pollard demonstrated that incorporating genomewide caRNA-DNA contact data in the deep learning model led to a more accurate recapitulation of 3D genome organization than using DNA sequence alone, supporting the idea that caRNA has widespread impacts on 3D genome organization. Furthermore, Kuang and Pollard demonstrated that trans-caRNAs, the caRNAs attached to the target genomic regions far away from the caRNA's transcription site, are particularly informative in recapitulating the 3D genome organization. This result holds even when the deep learning model has already taken chromatin accessibility information into account, revealing that caRNA-DNA contacts contain additional information about 3D chromatin features beyond chromatin accessibility. Based on AkitaR and in situ Mapping of RNA-Genome Interactome (iMARGI)-derived RNA-DNA contacts, Kuang and Pollard identify candidate R-loops formed by trans-caRNA and target genomic sequences, which frequently appear at loop anchors and stable TAD boundaries and are enriched with Alu transposons, alluding to a model where trans-caRNA-mediated R-loops promote chromatin looping and TAD boundary formation. These results demonstrate AkiraR's ability to generate hypotheses with deep mechanistic insights.

I recommend a speedy publication of this important work.

Response: We thank the reviewer for the encouraging comments.

Comment 2: The authors have cited two highly relevant iMARGI papers. Perhaps mentioning MARGI (Sridhar et al. Current Biology 2017 PMID: 28132817) in the Introduction and citing the original iMARGI paper (Yan et al. PNAS 2019 PMID: 30718424) could enrich the background information.

Response: We appreciate the reviewer's suggestion and have added MARGI in the Introduction section (line 44 and 72) and cited the original iMARGI paper (line 73).

Comment 3: Line 350: should "not other models in Fig. 4C could form ..." read "no other models in Fig. 4C could form"?

Response: We thank the reviewer for the suggestion and have changed "not other models" to "no other models" (line 351) to emphasize that the cell-type-specific interaction was not captured by any of the other models in Fig. 4c except the models incorporating *trans*-located caRNAs.

Response to Review 2:

Comment 1: This paper explores the role of chromatin-associated RNA (caRNA) in the spatial chromatin architecture. In brief, it shows that caRNAs likely play a causal role in establishing tissue-specific chromatin architecture, and identifies a substantial number of novel RNA genes that are likely involved in this process.

Previously it was shown, by the authors and others, that chromatin contact maps can be predicted with reasonable accuracy from DNA sequence alone. Here the authors show, first, that caRNAs are associated with intranuclear spatial position, open chromatin, and TAD boundaries, and second that in a deep learning model (AkitaR), caRNAs improve the prediction of chromatin contact maps when used as input, compared to models that use only DNA sequence, or a combination of DNA sequence and chromatin accessibility data.

The authors argue that this is likely because caRNA plays a causal role in chromatin architecture. Although the improvement in predictive ability is modest, in comparison with a model that uses both DNA sequence and chromatin accessibility data, the authors reasonably suggest (l. 384-5) that this might be the result of the chromatin accessibility signal partly explaining away a causal signal from caRNA, which could happen if caRNA interaction result in chromatin becoming accessible. In favor of this explanation, the authors show that the caRNA signal is highly specific to certain genes; in particular, the 10 genes that contributed most strongly to the model's predictive ability include 5 with known roles in chromatin structure.

I also liked the in-silico experiment, of taking random genomic loci and adding convergent CTCF motifs which are known to be associated with isolation domains and reduced contact frequency, and trans caRNAs of which the model's importance scores indicated an association with reduced contract frequency. The model indeed subsequently predicted reduced contact frequency for both types of synthetic sequence, suggesting a causal role. This is precisely the kind of in-silico experiments that are enabled by black-box machine learning models trained on extensive and complex data sets. Although the results of these experiments are never fully

conclusive as they are based on observational data and confounding remains possible, nevertheless it is a good test, as it is easy to interpret, and the results are what you would expect if the causal relationship indeed exists.

Response: We thank the reviewer for the insightful comments and valuable feedback.

Comment 2: The observation that the model that uses DNA sequence and caRNA as input performs best (Fig 3b) is key to the paper. Please therefore add estimates of standard errors in the mean (or median) to the figure to indicate how precise these performance measures are. This probably involves re-training the model several times.

Response: We appreciate the reviewer's suggestion. We added error bars (mean \pm standard error of the mean) to the bar plots in Fig. 3b, Fig. 4a, Supplementary Fig. 4a, Supplementary Fig. 8a and 9a. To estimate the variability, we retrained each model four more times to get five total models for computing the mean and standard error of the mean of each metric for every model.

We also calculated P-values for a two-sided Mann-Whitney U test of no difference in performance between each pair of models. The resulting U statistics and p-values are listed in a new Supplementary Table (Supplementary Table 2). Most comparisons were statistically significant. Denoting non-significant comparisons fit better in the space on the plot and was more visually readable (versus showing all the significant pairwise comparisons), but we are open to changing it if this is not interpretable to the reviewer.

Comment 3: I believe the argument referred to in the 2nd paragraph above, and mentioned in l. 384-5, should be expanded a bit in the discussion, as I think this is a crucial point. Even if adding caRNAs to the model does not significantly improve contact map predictions, the fact that this model uses known RNAs to make those predictions, and the likely possibility that caRNA interactions result in accessible DNA, make the model useful as an explanatory rather than just a predictive model. The authors do argue this point, but only briefly (l. 384-5); I would suggest they expand on this part a little.

Response: We thank the reviewer for this great suggestion. We have expanded the discussion to include the fact that the improvement from adding caRNAs is statistically significant alongside our point that many prioritized caRNAs are known to play roles in chromatin organization. We also elaborated on the point that *trans*-located caRNAs might promote and maintain open chromatin structures with different mechanisms.

Comment 4: Selecting trans caRNAs with high importance scores makes the in-silico experiment (l. 259-69) is somewhat circular. I suggest to re-do this experiment with RNA genes that were previously known to be involved with regulation of chromatin structure as well as expressed in HFFc6, including the 5 genes mentioned in l. 312.

Response: We agree that the in-silico experiment for examining whether trans-located caRNAs are able to increase insulation strength at TAD boundaries using trans-located caRNA signals with large negative scores is a bit circular. To address the reviewer's suggestion, we redid the simulation by selecting regions with relatively simple structures from the test dataset and then gradually increasing the *trans*-located caRNA signals of each type (e.g., lncRNA only) at TAD boundaries. For most RNA types, we observed that increasing the caRNAs signals increased TAD insulation. This included lncRNAs, which are the biotypes of many RNA genes that are known to be involved in 3D genome organization.

In addition, we realize that there was some confusion regarding how we did the prior in-silico experiment. Specifically, the importance scores were identified for caRNA features, which were the trans-located RNA signals at each genomic bin of 2,048 bp. Since the trans-located RNA signals represent the accumulated trans-located RNAs transcribed from all RNA genes that are at least 1Mb away, we were not able to identify the importance score of each RNA gene. Hence, we did not add specific caRNAs in the simulation, but rather used the cumulative caRNA value for bins with high importance scores. We replaced random bins' cumulative values with these values. This is probably less circular than what the reviewer thought we did. Nonetheless, we replaced this result with the new simulation which avoids circularity completely. We also revised the text to more clearly explain these in-silico experiments.

Comment 5: Thank you for the detailed data access information, and for making the code for exploratory data analysis and figures available. For full replicability and for a detailed understanding of the model, the modified AkitaR code is also required. Please also make this available in the GitHub repository.

Response: To address the reviewer's great suggestion, we have updated the GitHub repository with the modified AkitaR code. It also includes detailed instructions about how to train and evaluate AkitaR models.

Comment 6: Fig 5b, Supp Figure 14: please use the same order in the legend as in the plot. Please state whether the numbers in the heat map refer to 2log, 10log or natural log.

Response: Thank you for the feedback. We have reordered the legend and added explicit statements that the heatmap displayed \log_2 enrichment scores in both the plots and figure legends.

Comment 7: Supplementary Table 2, typo in the title.

Response: Thank you for catching the typo. We have fixed it.

Comment 8: l. 339 - pairwise2, a typo? Do you mean mutual best hit?

Response: Thank you for highlighting the ambiguity. Pairwise 2 is a python module in the Biopython package aimed for identifying global and local alignments between two sequences. Here, we identified the subsequences that exhibited the best alignment using the local alignment. We have revised the text (line 340-341) in the manuscript to address the confusion.

Response to reviewers' comments:

We would like to thank the reviewers for their valuable comments and helpful suggestions. We have addressed reviewers' comments below with our responses in **blue font** and made changes in the manuscript. To follow the formatting instructions and policy requirements, we have also made changes in all Figures to put the text in sentence case and implement other requested changes to the text and Figure legends. All changes in the main or supplemental text are highlighted in red.

Response to Review 1:

Comment 1: The chromatin is richly decorated with many chromatin-associated RNAs (caRNAs), begging the question of whether caRNA has widespread impacts on 3D genome organization. Kuang and Pollard developed a deep learning model AkitaR that jointly uses genomic sequence and the caRNAs attached to every genomic region to predict the 3D genome organization. Using AkitaR, Kuang and Pollard demonstrated that incorporating genomewide caRNA-DNA contact data in the deep learning model led to a more accurate recapitulation of 3D genome organization than using DNA sequence alone, supporting the idea that caRNA has widespread impacts on 3D genome organization. Furthermore, Kuang and Pollard demonstrated that trans-caRNAs, the caRNAs attached to the target genomic regions far away from the caRNA's transcription site, are particularly informative in recapitulating the 3D genome organization. This result holds even when the deep learning model has already taken chromatin accessibility information into account, revealing that caRNA-DNA contacts contain additional information about 3D chromatin features beyond chromatin accessibility. Based on AkitaR and in situ Mapping of RNA-Genome Interactome (iMARGI)-derived RNA-DNA contacts, Kuang and Pollard identify candidate R-loops formed by trans-caRNA and target genomic sequences, which frequently appear at loop anchors and stable TAD boundaries and are enriched with Alu transposons, alluding to a model where trans-caRNA-mediated R-loops promote chromatin looping and TAD boundary formation. These results demonstrate AkiraR's ability to generate hypotheses with deep mechanistic insights.

I recommend a speedy publication of this important work.

Response: We thank the reviewer for the encouraging comments.

Comment 2: The authors have cited two highly relevant iMARGI papers. Perhaps mentioning MARGI (Sridhar et al. Current Biology 2017 PMID: 28132817) in the Introduction and citing the original iMARGI paper (Yan et al. PNAS 2019 PMID: 30718424) could enrich the background information.

Response: We appreciate the reviewer's suggestion and have added MARGI in the Introduction section (line 44 and 72) and cited the original iMARGI paper (line 73).

Comment 3: Line 350: should "not other models in Fig. 4C could form ..." read "no other models in Fig. 4C could form"?

Response: We thank the reviewer for the suggestion and have changed "not other models" to "no other models" (line 351) to emphasize that the cell-type-specific interaction was not captured by any of the other models in Fig. 4c except the models incorporating *trans*-located caRNAs.

Response to Review 2:

Comment 1: This paper explores the role of chromatin-associated RNA (caRNA) in the spatial chromatin architecture. In brief, it shows that caRNAs likely play a causal role in establishing tissue-specific chromatin architecture, and identifies a substantial number of novel RNA genes that are likely involved in this process.

Previously it was shown, by the authors and others, that chromatin contact maps can be predicted with reasonable accuracy from DNA sequence alone. Here the authors show, first, that caRNAs are associated with intranuclear spatial position, open chromatin, and TAD boundaries, and second that in a deep learning model (AkitaR), caRNAs improve the prediction of chromatin contact maps when used as input, compared to models that use only DNA sequence, or a combination of DNA sequence and chromatin accessibility data.

The authors argue that this is likely because caRNA plays a causal role in chromatin architecture. Although the improvement in predictive ability is modest, in comparison with a model that uses both DNA sequence and chromatin accessibility data, the authors reasonably suggest (l. 384-5) that this might be the result of the chromatin accessibility signal partly explaining away a causal signal from caRNA, which could happen if caRNA interaction result in chromatin becoming accessible. In favor of this explanation, the authors show that the caRNA signal is highly specific to certain genes; in particular, the 10 genes that contributed most strongly to the model's predictive ability include 5 with known roles in chromatin structure.

I also liked the in-silico experiment, of taking random genomic loci and adding convergent CTCF motifs which are known to be associated with isolation domains and reduced contact frequency, and trans caRNAs of which the model's importance scores indicated an association with reduced contract frequency. The model indeed subsequently predicted reduced contact frequency for both types of synthetic sequence, suggesting a causal role. This is precisely the kind of in-silico experiments that are enabled by black-box machine learning models trained on extensive and complex data sets. Although the results of these experiments are never fully

conclusive as they are based on observational data and confounding remains possible, nevertheless it is a good test, as it is easy to interpret, and the results are what you would expect if the causal relationship indeed exists.

Response: We thank the reviewer for the insightful comments and valuable feedback.

Comment 2: The observation that the model that uses DNA sequence and caRNA as input performs best (Fig 3b) is key to the paper. Please therefore add estimates of standard errors in the mean (or median) to the figure to indicate how precise these performance measures are. This probably involves re-training the model several times.

Response: We appreciate the reviewer's suggestion. We added error bars (mean \pm standard error of the mean) to the bar plots in Fig. 3b, Fig. 4a, Supplementary Fig. 4a, Supplementary Fig. 8a and 9a. To estimate the variability, we retrained each model four more times to get five total models for computing the mean and standard error of the mean of each metric for every model.

We also calculated P-values for a two-sided Mann-Whitney U test of no difference in performance between each pair of models. The resulting U statistics and p-values are listed in a new Supplementary Table (Supplementary Table 2). Most comparisons were statistically significant. Denoting non-significant comparisons fit better in the space on the plot and was more visually readable (versus showing all the significant pairwise comparisons), but we are open to changing it if this is not interpretable to the reviewer.

Comment 3: I believe the argument referred to in the 2nd paragraph above, and mentioned in l. 384-5, should be expanded a bit in the discussion, as I think this is a crucial point. Even if adding caRNAs to the model does not significantly improve contact map predictions, the fact that this model uses known RNAs to make those predictions, and the likely possibility that caRNA interactions result in accessible DNA, make the model useful as an explanatory rather than just a predictive model. The authors do argue this point, but only briefly (l. 384-5); I would suggest they expand on this part a little.

Response: We thank the reviewer for this great suggestion. We have expanded the discussion to include the fact that the improvement from adding caRNAs is statistically significant alongside our point that many prioritized caRNAs are known to play roles in chromatin organization. We also elaborated on the point that *trans*-located caRNAs might promote and maintain open chromatin structures with different mechanisms.

Comment 4: Selecting trans caRNAs with high importance scores makes the in-silico experiment (l. 259-69) is somewhat circular. I suggest to re-do this experiment with RNA genes that were previously known to be involved with regulation of chromatin structure as well as expressed in HFFc6, including the 5 genes mentioned in l. 312.

Response: We agree that the in-silico experiment for examining whether trans-located caRNAs are able to increase insulation strength at TAD boundaries using trans-located caRNA signals with large negative scores is a bit circular. To address the reviewer's suggestion, we redid the simulation by selecting regions with relatively simple structures from the test dataset and then gradually increasing the *trans*-located caRNA signals of each type (e.g., lncRNA only) at TAD boundaries. For most RNA types, we observed that increasing the caRNAs signals increased TAD insulation. This included lncRNAs, which are the biotypes of many RNA genes that are known to be involved in 3D genome organization.

In addition, we realize that there was some confusion regarding how we did the prior in-silico experiment. Specifically, the importance scores were identified for caRNA features, which were the trans-located RNA signals at each genomic bin of 2,048 bp. Since the trans-located RNA signals represent the accumulated trans-located RNAs transcribed from all RNA genes that are at least 1Mb away, we were not able to identify the importance score of each RNA gene. Hence, we did not add specific caRNAs in the simulation, but rather used the cumulative caRNA value for bins with high importance scores. We replaced random bins' cumulative values with these values. This is probably less circular than what the reviewer thought we did. Nonetheless, we replaced this result with the new simulation which avoids circularity completely. We also revised the text to more clearly explain these in-silico experiments.

Comment 5: Thank you for the detailed data access information, and for making the code for exploratory data analysis and figures available. For full replicability and for a detailed understanding of the model, the modified AkitaR code is also required. Please also make this available in the GitHub repository.

Response: To address the reviewer's great suggestion, we have updated the GitHub repository with the modified AkitaR code. It also includes detailed instructions about how to train and evaluate AkitaR models.

Comment 6: Fig 5b, Supp Figure 14: please use the same order in the legend as in the plot. Please state whether the numbers in the heat map refer to 2log, 10log or natural log.

Response: Thank you for the feedback. We have reordered the legend and added explicit statements that the heatmap displayed \log_2 enrichment scores in both the plots and figure legends.

Comment 7: Supplementary Table 2, typo in the title.

Response: Thank you for catching the typo. We have fixed it.

Comment 8: l. 339 - pairwise2, a typo? Do you mean mutual best hit?

Response: Thank you for highlighting the ambiguity. Pairwise 2 is a python module in the Biopython package aimed for identifying global and local alignments between two sequences. Here, we identified the subsequences that exhibited the best alignment using the local alignment. We have revised the text (line 340-341) in the manuscript to address the confusion.